# Numerical Analysis of Crack Path Effects on the Vibration Behaviour of Aluminium Alloy Beams and Its Identification via Artificial Neural Networks

**DOI:** 10.3390/s25030838

**Published:** 2025-01-30

**Authors:** Hilal Doğanay Katı, Jamilu Buhari, Arturo Francese, Feiyang He, Muhammad Khan

**Affiliations:** 1Center for Life-Cycle Engineering and Management, Faculty of Engineering and Applied Sciences, Cranfield University, Cranfield MK43 0AL, UK; hilal.doganay@cranfield.ac.uk (H.D.K.); jamilu.buhari@shell.com (J.B.); a.francese@cranfield.ac.uk (A.F.); 2Faculty of Engineering and Natural Sciences, Department of Mechanical Engineering, Bursa Technical University, 16310 Bursa, Turkey

**Keywords:** crack identification, crack path, natural frequency and amplitude, artificial neural networks (ANNs)

## Abstract

Understanding and predicting the behaviour of fatigue cracks are essential for ensuring safety, optimising maintenance strategies, and extending the lifespan of critical components in industries such as aerospace, automotive, civil engineering and energy. Traditional methods using vibration-based dynamic responses have provided effective tools for crack detection but often fail to predict crack propagation paths accurately. This study focuses on identifying crack propagation paths in an aluminium alloy 2024-T42 cantilever beam using dynamic response through numerical simulations and artificial neural networks (ANNs). A unified damping ratio of the specimens was measured using an ICP^®^ accelerometer vibration sensor for the numerical simulation. Through systematic investigation of 46 crack paths of varying depths and orientations, it was observed that the crack propagation path significantly influenced the beam’s natural frequencies and resonance amplitudes. The results indicated a decreasing frequency trend and an increasing amplitude trend as the propagation angle changed from vertical to inclined. A similar trend was observed when the crack path changed from a predominantly vertical orientation to a more complex path with varying angles. Using ANNs, a model was developed to predict natural frequencies and amplitudes from the given crack paths, achieving a high accuracy with a mean absolute percentage error of 1.564%.

## 1. Introduction

Fatigue is the most common cause of fracture in metallic materials and contributes to over 80% of in-service failures in structural materials [1]. Factors such as material imperfections, inclusions, impurities, and operational conditions of machinery lead to crack initiation and propagation [2]. Untreated micro-cracks can break beam integrity, potentially resulting in catastrophic fatigue failures like the Aloha Airlines incident [3]. Given the serious consequences of the failures caused by crack propagation, researchers are compelled to seek methods to identify crack damage. Academia and industry have achieved damage identification with a lot of studies using structural vibration response [3,4,5]. The fundamental concept of this approach is that cracks change the material’s local structures, inducing a stiffness change, resulting in variations in the structure’s dynamic response or modal parameters [6,7].

Several review studies have been conducted in vibration-based structural health monitoring areas [8]. Das et al. [9] reviewed different vibration-based damage detection methods, including fundamental modal examination, the local diagnostic method, non-probabilistic methodology and the time series method. Among them, fundamental modal examination uses modal parameters like natural frequencies, mode shapes, and damping ratios for damage detection, and several research studies have been carried out. Detecting a single crack in structural components was one of the earliest problems tackled in the field. A two-step method for damage detection using natural frequencies was proposed in another study. It accurately localised and quantified damage through numerical simulations and experiments on cracked beams. However, a potential constraint of this approach is its tendency to symmetrically identify potential damage locations due to sole reliance on natural frequencies [10]. Likewise, Zai et al. [11] predicted crack depth in an aluminium 2024 cantilever beam operating at modal frequency. Based on the dynamic response parameters influenced by stiffness variations, the proposed method forecasted the crack depth under in situ conditions and thermo-mechanical loads. Similar work has been performed by Elshamy et al. [12]. They experimentally measured the modal frequency and mode shapes of a cracked cantilever beam with a tip probe and validated the test results by numerical simulation.

To address more complex scenarios, Altunışık et al. [13] identified modal parameters and detected cracks for multiple-cracked cantilever beams. They applied finite element (FE) models in ANSYS software for numerical analysis and conducted ambient vibration tests to extract dynamic characteristics. However, ambient vibration tests may not always provide controlled conditions for accurate measurements, causing potential errors in the results. Sanchez et al. [14] extended the damage detection to more complex geometries. A disk-like structure was investigated through modal response changes due to crack growth. Another challenge in crack detection is posed by breathing cracks, where the crack opens and closes under cyclic loading, as explored by Long et al. [15]. They developed a unique stiffness matrix for 3D finite element modelling of beams with breathing cracks to study crack effects on beam natural frequencies under bidirectional excitation, finding the impact on bending modes aligned with crack propagation direction. While the above studies used traditional methods, Alves et al. proposed a method to detect, locate and quantify multiple damages through an iterative FE model using genetic algorithms for a real railway bridge. However, this application on large-scale structures may be challenging, as significant damage may be necessary to observe noticeable frequency changes in addition to environmental factors which can influence natural frequencies, potentially causing false-positive indications of damage [16].

While much of the research has focused on detecting cracks, less attention has been given to predicting crack propagation paths, particularly using dynamic responses. A crack path reveals how the crack interacts with the surrounding material. The growth of cracks in different locations and directions has different effects on the strength of the structure. The identification of the path informs more effective repair strategies by determining the areas that require reinforcement.

Although not much research has been conducted in this area, a few studies have primarily aimed at predicting future crack paths based on stress distribution rather than directly using dynamic data. Alshoaibi et al. [17] used ANSYS to develop a model to predict crack propagation paths and fatigue crack growth using stress and implemented a mixed-mode fatigue life assessment. The maximum tangential stress theory was applied to determine the angle of crack growth. Chen et al. [18] introduced a numerical method using XFEM to accurately simulate mixed-mode crack growth path and fatigue life calculation for a PMMA beam specimen. Using Taguchi statistical analysis, Saber et al. [19] investigated the effects of cutouts on crack path and fatigue life in steel plates. Barter et al. [20] studied the effect of loading history on the crack path in aluminium alloy 7050-T7451. Specimens were subjected to a sequence consisting of four cycles of constant and variable amplitude loading. They generated corresponding different small crack growth rate data to aid understanding of fatigue crack growth mechanisms, with a focus on crack growth retardation and acceleration. However, in real-life scenarios, the stress-based method will encounter some challenges, such as sensor installation, making it difficult to measure stress. This method also neglected the crack closure effect, which can lead to inaccurate identification and prediction under complex scenarios. To address complex cases, Pierson et al. [21] employed a CNN to forecast the three-dimensional crack evolution in a polycrystalline alloy. They established spatial relationships between microstructural features and crack paths considering the vertical deviation (z-offset) of cracks along a specified axis. Similarly, Shen et al. [22] carried out a study and proposed a neural network model with Bayesian optimisation to predict short crack growth paths in α titanium alloy.

Most existing studies in vibration-based crack path identification area only reached the inclined crack investigation. Srivastava and Sethuraman [23] studied the effect of an inclined crack on the natural frequency of a beam using a strain energy equation derived from the Bernoulli–Euler beam theory and numerical model. Ma et al. [24] analysed the effects of crack propagation paths on time-varying mesh stiffness (TVMS) and the vibration responses of a perforated gear system using FE and lumped mass models (LMM). Their findings indicated that TVMS decreases with crack depth, with cracks propagating through the rim having a greater effect on system vibration than through the tooth under equal crack depth. Furthermore, cracks propagating in the rim direction had a more significant impact on vibration responses than those in the tooth direction. In addition, Yang et al. [25] proposed an empirical model to correlate the inclined crack angle and torsional spring stiffness used to determine the dynamic response for cantilever beams. Their results showed a decrease in dynamic response parameters with increased crack angle. Similarly, Francese et al. [26] studied the effect of surface crack orientation on the dynamic response of a hybrid material composed of 3D-printed ABS skin and an aluminium alloy 2014-T615 stiffener under mechanical loading. The results showed that the specimen with a 45° crack exhibited the lowest fundamental frequency compared to the ones with 0° and 30° cracks. Additionally, the crack propagation path was found to be greatly influenced by the crack orientation, with the 0° cracked sample having a linear propagation path and the 30°and 45° crack orientation samples having nonlinear propagation paths, respectively.

Other factors, such as crack branching, merging and irregular patterns, can complicate accurate crack path identification using existing models, particularly in dynamic and complex fracture scenarios [27]. Additionally, challenges such as directional mesh-bias sensitivity in finite element solutions, the high computational costs of numerical strategies and physical challenges in understanding phenomena such as multiple crack nucleation, crack opening/closing and crack intersection also exist [28].

The application of ANNs in crack propagation analysis has gained significant attention due to their ability to handle nonlinear problems, learn from data and provide efficient solutions for complex fracture mechanics challenges. According to Montalvão et al. [29], using ANNs for damage detection is highly recommended due to the complex nature of structural systems, where damage may arise in various locations. In the literature, Zang and Imregun [30] addressed the detection of structural damage by utilising measured frequency response functions (FRFs) as input data for ANNs. Szewczyk and Hajela [31] introduced a method for detecting structural damage by framing it as an inverse problem and solving it using neural networks. This approach effectively identified both the location and severity of damage within structural systems. In another study [32], an ANN was used to predict cracks in reinforced concrete beams based on eight parameters, including dimensions, material properties, and stress factors. The ANN model demonstrated reliable crack width predictions, validated through testing and training, offering a practical tool for similar structural assessments. Similarly, the identification of crack location in beam-like structures using ANNs has been studied by Sahin and Shenoi [33], Thatoi and Jena [34], and Suresh et al. [35]. Nevertheless, none of the existing studies have applied ANNs to identify crack paths in structures based on their dynamic characteristics, such as frequency and amplitude.

After this review, we can conclude that only slight progress has been made in detecting crack angles, as well as predicting future crack paths using stress analysis and geometric considerations; there remains a notable gap in the application of dynamic responses to identify crack propagation paths directly. Although vibration-based methods heavily rely on the accuracy and quality of vibration data [36], advancements in sensor technology make it easier to measure vibration data in real-life scenarios. ANNs can also process the complex structural dynamic behaviours introduced by the crack path. Therefore, the presented work addresses the crack path identification challenge through a numerical simulation for an aluminium alloy 2024-T42 cantilever beam with various crack paths. One ANN model between the crack profile and modal parameters was proposed based on the numerical results and validated by comparing it to analytical models.

## 2. Materials and Methods

### 2.1. Specimen Description

A cantilever beam was chosen in this research due to its high vibration behaviour sensitivity to changes in its structure. The beam design also simulates real-world applications, like in aerospace and construction, where similar conditions and failures happen. The aluminium alloy 2024-T42, which is widely used in the aerospace industry, was selected as the beam material. Table 1 shows the material properties.

The beam’s thickness is 3.16 mm, its length is 200 mm (266 mm, including the clamped part), and its width is 25 mm. The beam shape and size were designed to allow maximum stress near the fixed end when subjected to dynamic loading, as shown in Figure 1.

### 2.2. Crack Profile Scheme Development

Different crack paths were designed for the cantilever beam to compare their dynamic response. The crack parameters were introduced at 3.16 mm from the fixed support end of the specimen, allowing maximum stress to be generated at the tip of the crack [26]. The crack width was defined as 0.2 mm.

Several assumptions were made so that the cracks could simulate the actual situation. The overall crack path was divided into five propagation steps. At each step, the crack only had three scenarios. It stopped propagation and extended vertically downward or along the inclined 45-degree point to the fixed end for a vertical distance of 0.5 mm. This assumption allowed the simulation scheme to cover almost all possible crack paths. Therefore, there were 46 different cracked beams with crack parameters (crack depth and propagation path/angle), as shown in Table 2 and Figure 2, in addition to one intact beam.

It is worth noting that all cracks in the scheme are not the actual cases. The current crack path scheme covers the common crack path propagation area under reverse bending conditions. They are made to simplify the numerical analysis while capturing representative cases of crack propagation [26,38].

Afterwards, CATIA was used to build CAD models of the specimens with the crack paths shown in Table 2. The crack path width was set to 0.2 mm to represent the real crack path. These models were then imported into ANSYS, where numerical analysis was conducted using the corresponding data obtained, recorded and processed.

### 2.3. Simulation Testing Procedures

The ANSYS Modal and Harmonic modules were used to obtain the natural frequencies and resonance amplitudes of the intact beam and the 46 different cracked beams. Boundary conditions were first defined with a fixed support end introduced on the larger-width end of the beam. Full constraints were applied to the top and bottom surfaces at the beam’s fixed end (66 mm × 54 mm area in Figure 1). As shown in Figure 3, after multiple iterations with different element sizes to ensure simulation result convergence, a 2 mm mesh was determined with 1mm local refinement along the critical faces and crack path to capture key geometrical features. Then, the modal analysis was performed with the first three modal frequencies (1st, 2nd and 3rd bending modes).

The harmonic response analysis simulation was then carried out to determine the resonance amplitude. The excitation force was applied at the free end of the beam, and the frequency response function (FRF) in terms of the amplitude was measured at the same location (Figure 3b).

An impact laboratory test was carried out for the damping ratio on three specimen geometries (intact beam, beam with a vertical crack, and beam with an inclined crack of 1mm crack width and depth). The experimental setup is shown in Figure 4. The specimen was fixed on a V55 shaker (Data Physics, Hailsham, UK). The AFG 1062 signal generator (Tektronix, Beaverton, OR, USA) excited it with an impact force. A lightweight ICP^®^ 352A21 accelerometer vibration sensor (PCB, Blaine, WA, USA) was fixed on the beam’s free end to capture its response. The sensor was connected to the PC through a DAQ NI9234 card and DAQ NI9174 chassis (National Instrument, Austin, TX, USA). The captured response data were recorded by DAQExpress for the further process. The accelerometer’s sensitivity and sample frequency were 9.82 mV/g and 25,600 Hz, respectively. The damping ratios were then calculated using the half-power method. Their results were 0.0169, 0.0231 and 0.0188, respectively, with no significant difference. Therefore, a damping ratio of 0.02 for the three scenarios was obtained and used in the harmonic response simulations.

Intact beam FRF curves were obtained from an analytical solution and an ANSYS simulation. Then, the results were compared to see the effect of the damping ratio. As is seen in Figure 5, the curves exhibit peaks at nearly the same frequencies. As is known, the primary factor influencing the amplitude of the FRF curve near the natural frequency is damping [39]. The fact that the peak points and the amplitudes at these points are nearly the same indicates that the damping in the numerical model was correctly selected.

### 2.4. Artificial Neural Network

Artificial neural networks (ANNs) are complex and multi-layered artificial intelligence algorithms inspired by the learning and decision-making processes of the human brain. At their core, these networks consist of artificial neurons that simulate nerve cells, allowing them to process data and acquire learning capabilities across numerous layers. As shown in Figure 6, a neural network generally consists of several layers: the input layer, hidden layer(s) and output layer. Each layer multiplies the information from the previous layer by weights, adds a bias and passes it through an activation function. Each neuron in the hidden layer takes input values and performs the following operation:(1)$$z^{\left(1\right)}=W^{\left(1\right)}x+b^{\left(1\right)}$$
where $W^{\left(1\right)},\;x,\;b^{\left(1\right)},\;z^{\left(1\right)}$ are the weight matrix of the hidden layer, the input vector, the bias vector of the hidden layer and the weighted sum calculated for the hidden layer.

Each neuron in the hidden layer applies an activation function f (e.g., sigmoid, ReLU, etc.) to the value $z^{\left(1\right)}$. Then, the activation values from the hidden layer are passed to the output layer, where a similar operation is performed.(2)$$a^{\left(1\right)}={f(z}^{\left(1\right)})$$(3)$$z^{\left(2\right)}=W^{\left(2\right)}a^{\left(1\right)}+b^{\left(2\right)}$$(4)$$y=f_{out}(z^{\left(2\right)})$$
where $W^{\left(2\right)},\;b^{\left(2\right)},\;f_{out}$ are the weight matrix, the bias and the activation function of the output layer, respectively.

The objective of this study was to develop a neural network model to predict output values (frequencies (F1, F2, F3) and amplitudes (A1, A2, A3)) based on the input data (crack paths) derived from a provided dataset. The initial step in the ANN process is the preparation of the dataset to accurately represent the crack paths in terms of their horizontal (dx) and vertical (dy) components. For instance, consider the crack path A1-B1-C2; the segment A-B has dy = 0.5 and dx = 0 as the path was vertical. In contrast, the segment B-C has both dy = 0.5 and dx = 0.5, reflecting a diagonal path at a 45° angle, while the other segments C-D, D-E and E-F all remain 0.

The implemented ANN approach used in this study is a feedforward neural network with ten inputs, one hidden layer with ten neurons, and six outputs. A feedforward neural network was constructed using the **fitnet** function from MATLAB’s Neural Network Toolbox. The activation functions used for the network were Logistic Sigmoid (**logsig**) and Linear (**purelin**) for the hidden layer and output layer functions, respectively. The data were divided into three sets: training, validation and testing. The split was 80% for training, 10% for validation and 10% for testing. The network was trained using the Levenburg–Marquardt (LM) backpropagation algorithm (**trainlm**), which is widely used, well suited for medium-sized datasets and provides efficient training by balancing between the gradient descent and Gauss–Newton methods. The training process aimed to minimise the mean squared error (MSE) between the predicted outputs and the target values. To address the stochastic nature of neural network training (due to the random initialisation of weights and biases), the network was trained 50 times. The predicted outputs from each run were accumulated, and an average output was calculated. This approach minimised the effect of outliers and provided a more reliable estimate of the model’s performance.

After analysing the training and testing dataset, the proposed ANN model was validated against a dataset that was not used for the training and testing dataset, which had five different crack paths in terms of the natural frequency results. Furthermore, these five different crack paths were analytically modelled for Euler–Bernoulli beams (for detailed information, see references [25,40]) and natural frequencies were obtained.

## 3. Results and Discussion

The simulation was conducted with the first three bending mode frequencies and the first mode amplitude. Results obtained through ANSYS are recorded in Table 3. The simulations focused on evaluating the impact of varying crack depths and propagation paths on the beam’s natural frequencies and amplitude.

### 3.1. Modal Behaviour Due to the Influence of Crack Depth

Figure 7 shows how the modal parameters change with increased crack depth. The average value of natural frequencies was calculated for different path profiles with the same crack depth. When the path is not considered and only the depth is considered, the beam’s natural frequency decreases significantly as the crack depth increases, which aligns with observations in existing studies [41,42,43,44,45].

For an intact beam, the first natural frequency was 64.657 Hz. However, as the crack depth grew from 0.5 mm to 2.5 mm, the average fundamental frequency kept decreasing, reaching 39.272 Hz at the deepest crack, (A1-B1-C1-D2-E3-F4) 2.5 mm. The second and third natural frequencies followed the same pattern. This happens because cracks reduce the structural local stiffness, which lowers its natural frequencies. In addition, it is worth noting that the natural frequencies tend to decrease faster as the crack depth increases. The reason behind this acceleration is the nonlinear decrease in structural stiffness with increasing crack depth. As for the reduction in the natural frequency of cracked beams, the first mode frequency shows the highest percentage drop, around 40%, when the crack depth reaches 2.5 mm. Despite the absolute difference between the intact beam’s 65 Hz and the 2.5mm cracked beam’s 40 Hz being less than the second and third mode frequency changes, 70 Hz and 150 Hz, the fundamental frequency is still the most sensitive modal parameter. This is due to the lower mode being characterised by a global deformation pattern that involves larger portions of the structure.

Meanwhile, the first mode amplitude at the beam tip demonstrates the opposite trend. Starting from approximately 14 mm under the intact condition, it increases with deeper cracks, reaching 30 mm when the crack depth is 2.5 mm (such as A1-B2-C3-D4-E5–F6). This is primarily because the beam responds more to vibrations as it becomes weaker.

### 3.2. Modal Behaviour Due to the Influence of Crack Angle

#### 3.2.1. Fully Inclined Crack Influence on Structural Dynamics

Numerous previous studies have considered the case of the modal behaviours of beams with different crack depths. Similarly, the crack angle effect has been extensively investigated [23,25,46]. Most studies consider a fully inclined crack and suggest that when the crack depth is constant, the greater angle of inclination of the crack can lower the natural frequency.

The simulation results confirm the same argument. As shown in Table 4, as the crack depth increases, the natural frequency consistently decreases for both crack orientations. Meanwhile, the inclined crack path exhibits a more significant reduction in natural frequency compared to the vertical crack path. For example, at a crack depth of 2.5 mm, the natural frequency for the vertical crack path A1 to F1 is 41.938 Hz, whereas it is significantly lower at 40.108 Hz for the inclined crack path A1 to F6. This suggests that inclined cracks have a more significant impact on structural stiffness due to their effect on both axial and shear stiffness.

In addition to frequency changes, the amplitude of the first mode increases as the crack depth grows, indicating greater flexibility and more vibrational responses. The amplitude growth is more significant for inclined cracks as well. For instance, as shown in Table 4, at a crack depth of 2.5 mm, the amplitude for the inclined crack path (A1-B2-C3-D4-E5-F6) is 30.897 mm compared to 28.708 mm for the vertical crack path (A1-B1-C1-D1-E1-F1). This difference can be attributed to the geometric orientation of inclined cracks, which amplifies modal displacements more effectively than vertical cracks.

The greater impact of inclined cracks on natural frequency and amplitude can be explained by their influence on both axial and shear stiffness, while vertical cracks primarily affect axial stiffness. Furthermore, as crack depth increases, the reduction in natural frequency becomes more nonlinear, with a sharper decline observed at greater depths. This is due to the expanding area of stiffness degradation as the crack propagation becomes deeper.

#### 3.2.2. Locally Inclined Crack Influence on Structural Dynamics

While fully inclined crack propagation affects both the frequency and amplitude of the beam, a locally inclined segment in a crack path exhibits similar effects. When the crack angle changes from vertical to inclined, the natural frequencies decrease and the amplitudes increase, even at the same crack depth. For example, as shown in Table 3, at a 2.0 mm crack depth, the crack path A1-B1-C1-D1-E1 with vertical crack propagation (D1-E1) lowers the frequency to 55.047 Hz, while the crack path A1-B1-C1-D1-E2 with inclined propagation (D1-E2) reduces it further to 53.930 Hz, with a slight increase in amplitude from 17.940 mm to 18.552 mm.

This happens because an inclined crack path segment reduces the beam’s local stiffness more, increases its flexibility, leads to higher vibration amplitudes and creates more complex stress distribution, leading to higher energy dissipation. This explains the changes in both frequency and as the crack angle changes. Similar trends were observed at all other crack depths (from 0.5 to 2.5 mm).

### 3.3. Crack Path Influence on Modal Behaviour

Additionally, the results showed that even when cracks have the same start and end points, the path they take can affect the structure’s natural frequency and amplitude. As illustrated in Figure 8, the simplest scenarios are the A1-B1-C2 and A1-B2-C2 crack paths. Table 5 shows that the deeper inclined segment between 0.5 and 1 mm crack depth has a more significant influence on modal frequency drop than the initial inclined path at the first 0.5 mm crack depth.

In the A1-B1-C2 crack path, the first mode frequency, 62.863 Hz, reduces by approximately 1.8 Hz, compared with the intact beam’s 64.657 Hz, while in the A1-B2-C2 crack path, the frequency only decreases by 1.5 Hz. The second- and third-order natural frequencies of the A1-B1-C2 crack path are even lower than that of A1-B2-C2 by 1.37 and 3.1 Hz, respectively.

Similar trends are observed in more complex crack paths as well. For example, at a crack depth of 2.5 mm, different crack paths from A1 to F2 exhibit varying results in frequency and amplitude, as shown in Table 6. As it is seen, the first mode frequency decreases as the inclined crack path segment shifts downward from B1-C2 to D1-E2, indicating that the structure becomes less stiff along the path A1-B1-C1-D1-E2-F2 with a corresponding increase in amplitude.

The reasons for this more significant frequency reduction and increased amplitude for a crack path with a deeper inclined segment can be attributed to the nonlinear effect of crack propagation on the structural stiffness. As the crack propagates deeper, the affected stiffness region increases nonlinearly, leading to greater degradation in both axial and shear stiffness. The inclined crack path segment further strengthens this effect by reducing stiffness in multiple directions, thereby amplifying their impact on modal behaviours.

Nevertheless, the influence becomes more complicated and irregular in some crack path conditions. One example is when the crack in Figure 7 continues to propagate to D2. As mentioned before, the shallower inclined crack path segment should lead to higher structural stiffness. In other words, the crack path A1-B2-C2-D2 should have had a higher modal frequency than the crack path A1-B1-C2-D2. However, the numerical results reveal the opposite trend. As shown in Table 5, although the differences are very small, the first mode frequency of the crack path A1-B2-C2-D2 is 60.459 Hz, which is lower than the 60.505 Hz of the crack path A1-B1-C2-D2 by approximately 0.05 Hz. In addition, the crack path A1-B1-C1-D1-E1-F2 in Table 6 also exhibits irregular behaviour compared to the other three crack paths. As shown in Figure 9, its first mode frequency reaches 41.709 Hz, which is even higher than the 41.685 Hz of the crack path A1-B1-C2-D2-E2-F2. This unusual behaviour still requires a more in-depth investigation.

During the development of the numerical mode, the analysis revealed that the behaviour of the data for the dx component of different crack paths was irregular compared to the dy component. This discrepancy arose because, unlike the dy direction where the load was directly applied in the y-direction in the ANSYS simulation, the dx direction was more influenced by secondary effects like lateral deformation and stress redistribution. The presence of cracks reduced stiffness, causing unpredictable shifts in the x-direction. Additionally, damping, which reduces vibrations, was less effective in the x-direction due to the indirect load application, resulting in more erratic dx data. Thus, the direction of the applied load played a significant role in the complex behaviour of the dx data compared to the dy data.

## 4. Modelling and Validation

An ANN model was trained to find natural frequencies for different crack paths. The trained model was used to identify/predict the crack propagation path for a new set of input conditions as a test. The new crack paths are defined in Table 7. The crack path end points considered steps other than 0.5 mm. In other words, the tested crack paths included segments with different inclined angles rather than just 45°. However, the cracks were designed for easy validation, allowing their first mode natural frequency to be calculated analytically.

In order to validate the ANN model, the natural frequencies were obtained analytically and from the ANN model. In the numerical simulation in Section 2, the inclined path segment orientation angles were always 45° for different paths. However, for the ANN model trained by these numerical modal frequencies, the validation results were very close to the analytical model in Table 7. Even if the crack has a different orientation angle (such as a1b2c3), the ANN model still gives accurate results with a 1.564% of the mean absolute percentage error (MAPE). The MAPE is a measure used to evaluate the accuracy of a prediction model by expressing the mean absolute error as a percentage of the actual values. It is calculated as Equation (5):(5)$$MAPE=\frac{1}{n}\sum_{i=1}^{n}\left|\frac{A_{i}-F_{i}}{A_{i}}\right|\times 100$$
where A and F represent, respectively, the actual and predicted values for observation i.

## 5. Conclusions

This research aimed to investigate the influence of crack characteristics, with a focus on different paths, on the vibration response (frequency and amplitude) of aluminium alloy 2024 T42 cantilever beams. Unlike prior research, which emphasised crack depth and angle, this work expands the scope to include the effects of whole crack paths with varying slopes and kink points.

From the numerical results obtained and analysis carried out, the crack propagation path critically impacts structural stiffness and vibration behaviour. Even cracks with the same start and end points can produce different dynamic responses depending on their propagation paths. Deeper inclined segments were found to increase frequency reductions and amplitude increases due to the nonlinear effects of crack propagation on structural stiffness. The ANN model effectively predicted natural frequencies. It filled the existing gaps in vibration-based crack path detection for structural health monitoring. Future work could focus on extending this approach to complex geometries and materials, further enhancing its applicability in diverse engineering fields.

## Figures and Tables

**Figure 1 sensors-25-00838-f001:**
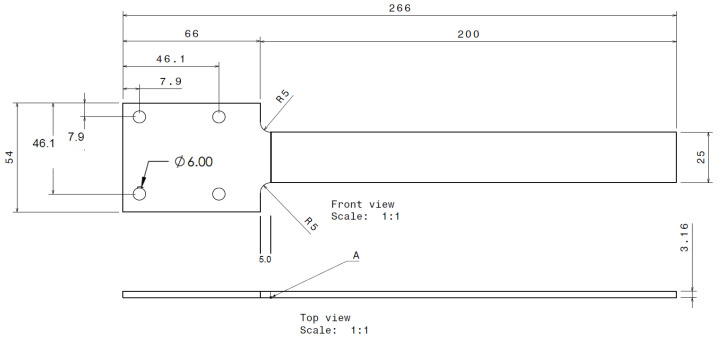
Specimen dimensions.

**Figure 2 sensors-25-00838-f002:**
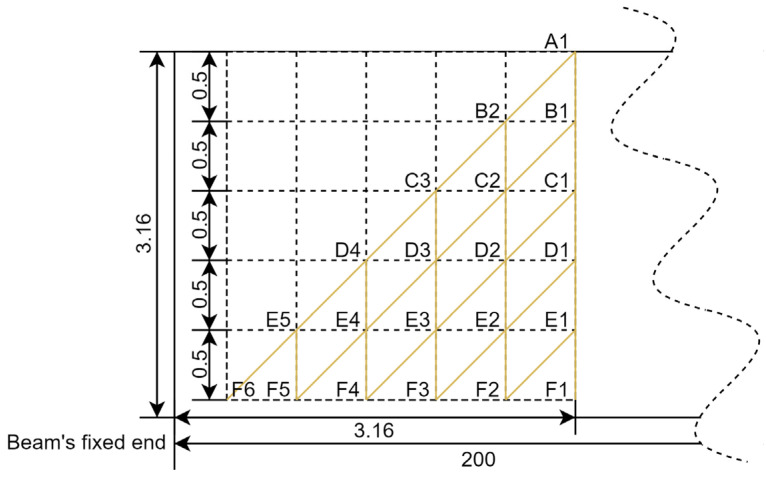
Crack path schematic. Yellow lines represent the possible crack paths.

**Figure 3 sensors-25-00838-f003:**
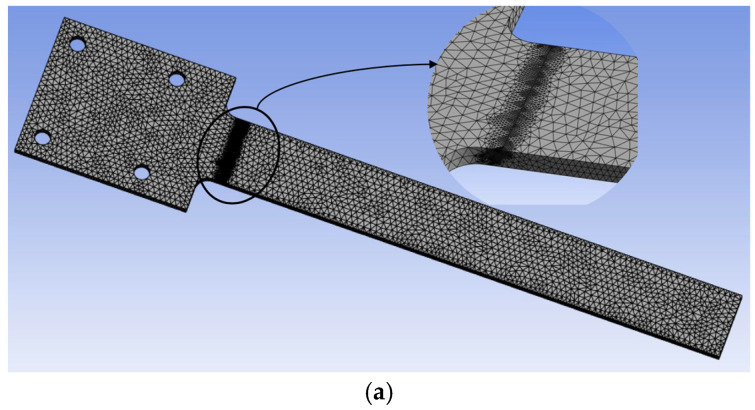
(**a**) Geometry meshing; (**b**) Red label and arrow line represent the amplitude measurement location and the applied force.

**Figure 4 sensors-25-00838-f004:**
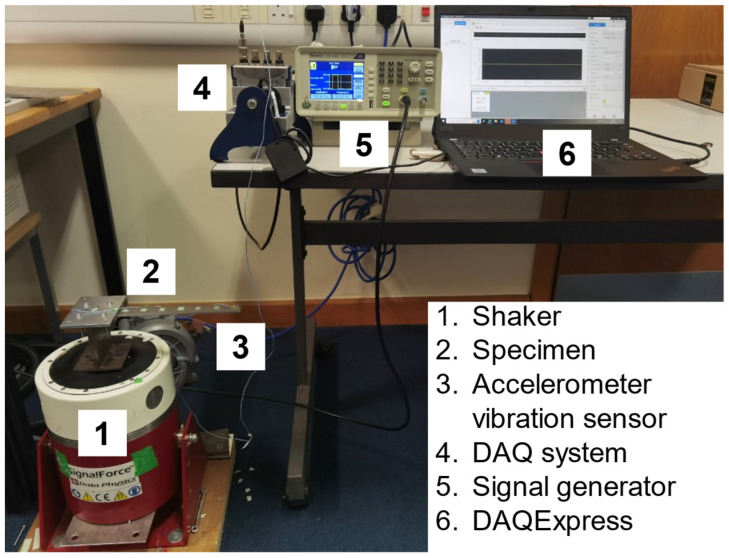
Experimental setup for the damping ratio measurement.

**Figure 5 sensors-25-00838-f005:**
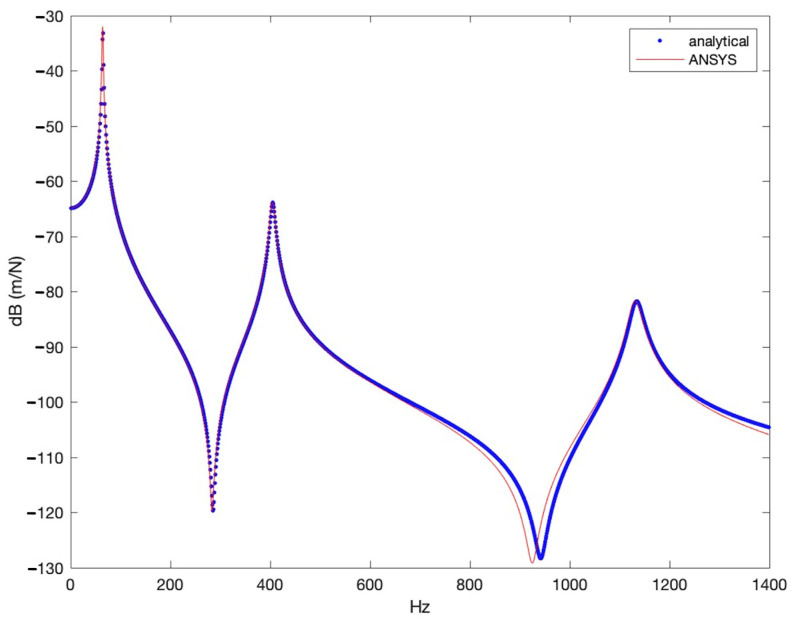
Comparison of FRF curves of the intact beam.

**Figure 6 sensors-25-00838-f006:**
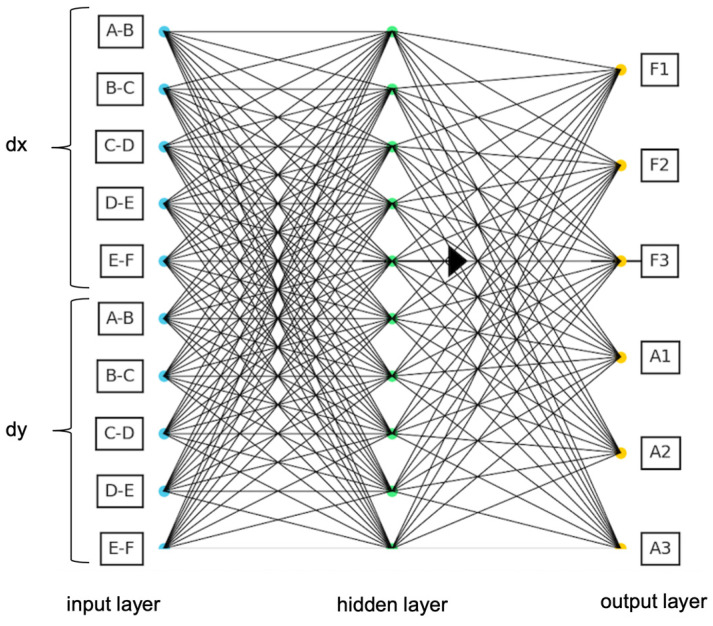
ANN architecture in a simplified form.

**Figure 7 sensors-25-00838-f007:**
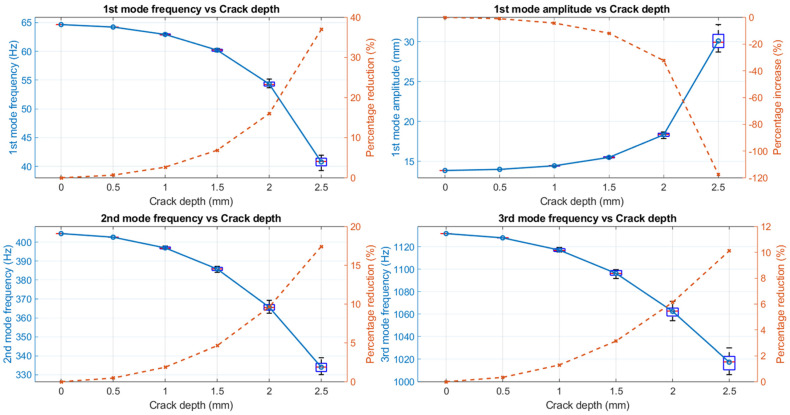
Modal frequency and amplitude change with the crack depth. Average values are calculated for different crack profiles with the same crack depth.

**Figure 8 sensors-25-00838-f008:**
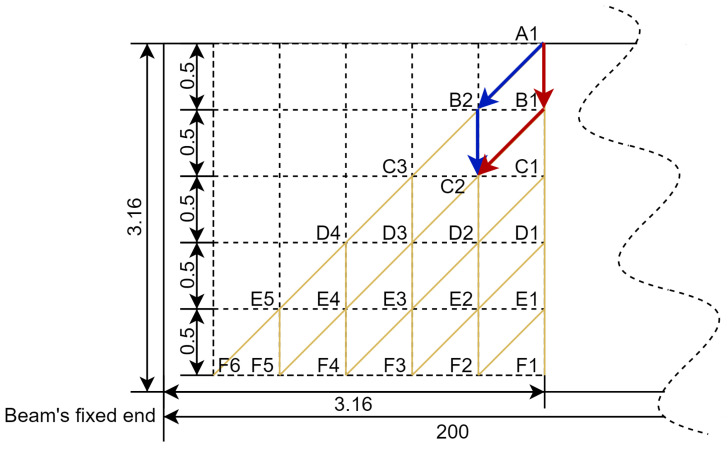
Two different crack paths with the same start–end point schematic.

**Figure 9 sensors-25-00838-f009:**
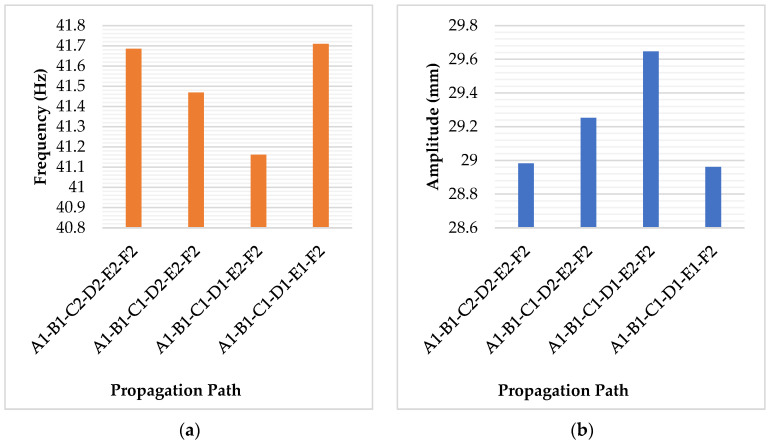
Inclined segment position influences the modal behaviour of the crack path ending at F2. (**a**). First mode frequency. (**b**). First mode amplitude.

**Table 1 sensors-25-00838-t001:** Aluminium alloy mechanical properties [37].

Material	Aluminium Alloy 2024-T42
Density	2770 kg/m^3^
Young’s Modulus	71 GPa
Poisson’s Ratio	0.33
Fatigue Strength	140 MPa
Shear Modulus	27 GPa
Shear Strength	280 MPa
Tensile Ultimate Strength	470 MPa
Tensile Yield Strength	300 MPa

**Table 2 sensors-25-00838-t002:** Simulation testing scheme.

S/No	Crack Path ID	Crack Vertical Depth (mm)	S/No	Crack Path ID	Crack Vertical Depth (mm)
1	A1-B1	0.5	24	A1-B2-C3-D3-E4	2.0
2	A1-B2	0.5	25	A1-B2-C3-D4-E4	2.0
3	A1-B1-C1	1.0	26	A1-B2-C3-D4-E5	2.0
4	A1-B1-C2	1.0	27	A1-B1-C1-D1-E1-F1	2.5
5	A1-B2-C2	1.0	28	A1-B1-C1-D1-E1-F2	2.5
6	A1-B2-C3	1.0	29	A1-B1-C1-D1-E2-F2	2.5
7	A1-B1-C1-D1	1.5	30	A1-B1-C1-D1-E2-F3	2.5
8	A1-B1-C1-D2	1.5	31	A1-B1-C1-D2-E2-F2	2.5
9	A1-B1-C2-D2	1.5	32	A1-B1-C1-D2-E2-F3	2.5
10	A1-B1-C2-D3	1.5	33	A1-B1-C1-D2-E3-F3	2.5
11	A1-B2-C2-D2	1.5	34	A1-B1-C1-D2-E3-F4	2.5
12	A1-B2-C2-D3	1.5	35	A1-B1-C2-D2-E2-F2	2.5
13	A1-B2-C3-D3	1.5	36	A1-B1-C2-D2-E2-F3	2.5
14	A1-B2-C3-D4	1.5	37	A1-B1-C2-D2-E3-F3	2.5
15	A1-B1-C1-D1-E1	2.0	38	A1-B1-C2-D2-E3-F4	2.5
16	A1-B1-C1-D1-E2	2.0	39	A1-B1-C2-D3-E4-F4	2.5
17	A1-B1-C1-D2-E2	2.0	40	A1-B1-C2-D3-E4-F5	2.5
18	A1-B1-C1-D2-E3	2.0	41	A1-B2-C3-D3-E3-F3	2.5
19	A1-B1-C2-D2-E2	2.0	42	A1-B2-C3-D3-E3-F4	2.5
20	A1-B1-C2-D2-E3	2.0	43	A1-B2-C3-D4-E4-F4	2.5
21	A1-B1-C2-D3-E3	2.0	44	A1-B2-C3-D4-E4-F5	2.5
22	A1-B1-C2-D3-E4	2.0	45	A1-B2-C3-D4-E5-F5	2.5
23	A1-B2-C3-D3-E3	2.0	46	A1-B2-C3-D4-E5-F6	2.5

**Table 3 sensors-25-00838-t003:** ANSYS simulation results.

Intact Beam Ansys Simulation Results	1st Mode Freq. (Hz)	1st Mode amplitude (mm)	2nd Mode Freq. (Hz)	3rd Mode Freq. (Hz)					
Intact Beam	64.657	13.844	404.540	1131.800					
Cracked Beams Modal and Harmonic Response Ansys Simulation Results									
Vertical Crack Propagation					Inclined Crack Propagation				
Crack Propagation Angle: 0°	Inflection Points (dx, dy):dx:0; dy:0.5				Crack Propagation Angle: 45°	Inflection Points (dx, dy):dx:0.5; dy:0.5			
Crack Path ID	1st Mode Freq. (Hz)	1st Mode amplitude (mm)	2nd Mode Freq. (Hz)	3rd Mode Freq. (Hz)	Crack Path ID	1st Mode Freq. (Hz)	1st Mode amplitude (mm)	2nd Mode Freq. (Hz)	3rd Mode Freq. (Hz)
Crack Depth: 0.5 mm									
A1-B1	64.236	13.985	402.670	1128.100	A1-B2	64.228	13.987	402.590	1127.900
Crack Depth: 1.0mm									
A1-B1-C1	62.916	14.448	396.960	1117.300	A1-B1-C2	62.863	14.466	396.590	1116.300
A1-B2-C2	63.146	14.366	397.960	1119.400	A1-B2-C3	62.846	14.471	396.350	1115.400
Crack Depth: 1.5 mm									
A1-B1-C1-D1	60.241	15.483	386.420	1098.600	A1-B1-C1-D2	60.050	15.558	385.370	1095.900
A1-B1-C2-D2	60.505	15.372	387.210	1099.600	A1-B1-C2-D3	60.279	15.461	386.000	1096.600
A1-B2-C2-D2	60.459	15.391	386.990	1099.100	A1-B2-C2-D3	59.995	15.577	384.750	1093.900
A1-B2-C3-D3	60.251	15.471	385.700	1095.500	A1-B2-C3-D4	59.918	15.605	384.070	1091.800
Crack Depth: 2.0 mm									
A1-B1-C1-D1-E1	55.047	17.940	369.250	1071.400	A1-B1-C1-D1-E2	53.930	18.552	365.260	1063.600
A1-B1-C1-D2-E2	55.188	17.854	368.990	1069.400	A1-B1-C1-D2-E3	54.300	18.333	365.730	1063.000
A1-B1-C2-D2-E2	54.490	18.234	366.770	1065.800	A1-B1-C2-D2-E3	54.018	18.490	364.750	1061.100
A1-B1-C2-D3-E3	54.843	18.030	367.220	1065.000	A1-B1-C2-D3-E4	53.676	18.675	363.020	1056.800
A1-B2-C3-D3-E3	54.427	18.258	365.830	1062.500	A1-B2-C3-D3-E4	53.999	18.489	363.920	1058.100
A1-B2-C3-D4-E4	54.167	18.392	364.320	1058.500	A1-B2-C3-D4-E5	53.751	18.620	362.470	1054.200
Crack Depth: 2.5 mm									
A1-B1-C1-D1-E1-F1	41.938	28.708	339.010	1029.900	A1-B1-C1-D1-E1-F2	41.709	28.961	337.690	1026.400
A1-B1-C1-D1-E2-F2	41.162	29.646	336.480	1024.300	A1-B1-C1-D1-E2-F3	40.166	30.947	333.820	1018.800
A1-B1-C1-D2-E2-F2	41.469	29.253	336.980	1024.800	A1-B1-C1-D2-E2-F3	40.015	30.301	331.020	1009.900
A1-B1-C1-D2-E3-F3	41.043	29.760	335.080	1019.900	A1-B1-C1-D2-E3-F4	39.272	32.145	330.090	1009.300
A1-B1-C2-D2-E2-F2	41.685	28.982	337.360	1025.300	A1-B1-C2-D2-E2-F3	40.527	30.450	334.370	1019.400
A1-B1-C2-D2-E3-F3	41.118	29.662	335.190	1020.000	A1-B1-C2-D2-E3-F4	40.078	31.025	332.440	1014.500
A1-B1-C2-D3-E4-F4	40.881	29.931	333.590	1015.400	A1-B1-C2-D3-E4-F5	39.939	31.176	331.010	1010.000
A1-B2-C3-D3-E3-F3	41.425	29.268	335.690	1020.600	A1-B2-C3-D3-E3-F4	40.322	30.685	332.780	1014.700
A1-B2-C3-D4-E4-F4	41.403	29.256	334.450	1016.400	A1-B2-C3-D4-E4-F5	40.075	30.982	331.140	1010.000
A1-B2-C3-D4-E5-F5	40.842	29.941	332.300	1011.100	A1-B2-C3-D4-E5-F6	40.108	30.897	330.070	1006.200

**Table 4 sensors-25-00838-t004:** Vertical and inclined crack path.

Crack Depth	Vertical Crack Path	1st Mode Freq. (Hz)	1st Mode Amplitude (mm)	Inclined 45° Crack Path	1st Mode Freq. (Hz)	1st Mode Amplitude (mm)
0	-	64.657	13.844	-	-	-
0.5	A1-B1	64.236	13.985	A1-B2	64.228	13.987
1	A1-B1-C1	62.916	14.448	A1-B2-C3	62.846	14.471
1.5	A1-B1-C1-D1	60.241	15.483	A1-B2-C3-D4	59.918	15.605
2	A1-B1-C1-D1-E1	55.047	17.94	A1-B2-C3-D4-E5	53.751	18.62
2.5	A1-B1-C1-D1-E1-F1	41.938	28.708	A1-B2-C3-D4-E5-F6	40.108	30.897

**Table 5 sensors-25-00838-t005:** Numerical results for some crack paths with the same start and end points.

Crack Path Start/End Points	Crack Path ID	1st Mode Freq. (Hz)	1st Mode Amplitude (mm)	2nd Mode Freq. (Hz)	3rd Mode Freq. (Hz)	Crack Path ID	1st Mode Freq. (Hz)	1st Mode Amplitude (mm)	2nd Mode Freq. (Hz)	3rd Mode Freq. (Hz)
A1-C2	A1-B1-C2	62.863	14.466	396.590	1116.300	A1-B2-C2	63.146	14.366	397.960	1119.400
A1-C2-D2	A1-B1-C2-D2	60.505	15.372	387.210	1099.600	A1-B2-C2-D2	60.459	15.391	386.990	1099.100

**Table 6 sensors-25-00838-t006:** Example of crack paths with similar start and end points. Red colour represents the inclined segments in the crack path.

Crack Path	Start Point	End Point	1st mode Frequency (Hz)	Amplitude (mm)
A1-B1-C2-D2-E2-F2	A1	F2	41.685	28.982
A1-B1-C1-D2-E2-F2			41.469	29.253
A1-B1-C1-D1-E2-F2			41.162	29.646
A1-B1-C1-D1-E1-F2			41.709	28.961

**Table 7 sensors-25-00838-t007:** Validation of ANN model.

New Crack Paths	Crack Depth (mm)		Frequencies (Hz)						Error (%)		
	dx	dy	Analytical			ANN			E1	E2	E3
			1st	2nd	3rd	1st	2nd	3rd	1st	2nd	3rd
a1b1	0	0.25	64.477	404.160	1131.878	64.300	402.700	1127.900	0.275	0.361	0.351
a1b1c1	0	0.75	63.493	398.825	1118.826	63.000	398.500	1120.100	0.776	0.081	0.113
a1b2c3	0.5	0.86	63.102	396.627	1113.231	63.650	400.239	1122.129	0.860	0.902	0.792
a1b1c1d1	0	1.25	61.329	387.974	1093.970	62.100	394.200	1113.200	1.241	1.604	1.757
a1b1c1d1e1	0	1.75	57.658	372.004	1061.213	58.000	378.900	1086.000	0.593	1.853	2.335
a1b1c1d1e1f1	0	2.25	52.907	355.044	1030.827	47.300	351.100	1047.800	10.597	1.110	2.335

## Data Availability

Data are contained within the article.

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
