# Peer review of "Numerical Analysis of Crack Path Effects on the Vibration Behaviour of Aluminium Alloy Beams and Its Identification via Artificial Neural Networks"

_sensors, 2025, doi:10.3390/s25030838_

Round 1

Reviewer 1 Report

Comments and Suggestions for Authors

Review attached

Reviewer 2 Report

Comments and Suggestions for Authors

This study aims to investigate the fatigue crack growth behavior in aluminum alloy beams. The influence of crack depths and growth angles on the natural frequencies and resonance amplitudes was revealed based on the numerical simulations. Subsequently, the model was developed to predict natural frequencies and amplitudes from a given crack path. These investigations have considerable engineering significance. However, the manuscript lacks refinement and requires improvement. Please consider the following suggestions:

(1) There appears to be a mismatch between the title and the content of the paper. This study focuses on investigating the effects of crack depths and growth angles on the natural frequencies and resonance amplitudes of aluminum alloy beams.

(2) 2.2 Crack Profile Scheme Development”,Limiting the analysis to vertical downward and 45° inclined crack growth may overlook other potential growth paths, leading to inaccurate and incomplete conclusions.

(3) 2.3 Simulation Testing Procedures”,Please provide further clarification on the finite element model boundary conditions and how to introduce fatigue cracks.

(4) 4. Modelling and validation”,Please provide further clarification on the training process and details of the machine learning model. Additionally, the small validation set may lead to evaluation metrics in Table 7 being unreliable.

Comments on the Quality of English Language

The writting needs improvement.

Reviewer 3 Report

Comments and Suggestions for Authors

In this paper, numerical simulations and Artificial Neural Networks have been used to analyse the crack extension paths in cantilever beams with different inclination angles, to investigate the effect of different inclination depths on the amplitude and frequency, and to demonstrate that the ANN model can effectively predict the intrinsic frequency. The study is interesting and informative, but certain parts need to be improved before the manuscript is published.

(1)  “Page 1, Line 23-25”

“Using ANNs, a model was developed to predict natural frequencies and amplitudes from given crack paths, achieving a high accuracy with a mean absolute percentage error of less than 2%.”

The exact data need to be added.

(2)  “Page 2, Line 88-90’’

“Chen et al. [18] introduced a numerical method using XFEM to accurately simulate mixed-mode crack growth path and fatigue life calculation for a PMMA beam specimen.”

Please write the citation in a uniform format.

(3)  “Page 2, Line 91-92’’

“Barter et al. studied the effect of loading history on the crack path in aluminium alloy 7050-T7451.”

Please explain the meaning of loading history in the literature.

(4)  “Page 3, Line 146-147”

“Nevertheless, none of the existing studies have ap-plied ANNs to identify the crack path in the structures.”

In the study in Ref. 32, which used ANN to predict the size of cracks, etc., please explain the apparent difference from crack paths.

(5) “Page 5, Line 179-181”

“At each step, the crack will only have three scenarios. It stops propagation and extends vertically downward or along the inclined 45 degrees point to the fixed end for a vertical distance of 0.5 mm.”

Please explain the three scenarios of cracking.

(6) “Page 5, Line 182-184”

“There-fore, there are 46 different cracked beams with crack parameters (crack depth and propa-gation path/ angle), as shown in Table 2 and Figure 2, in addition to one intact beam.”

Please provide a brief description of the propagation pathways in Figure 2 and Table 2.

(7)  “Page 6, Line 203-204”

“The excitation force was applied at the free end of the beam and frequency response function (FRF) in terms of the amplitude was measured at the same location (Figure 4).”

Please make an analysis of the curve in Figure 4.

(8)  “Page 6, Line 204-206”

“For the damping ratio, an impact laboratory test was carried out on three specimen geometries (Intact beam, beam with vertical crack and beam with inclined crack of 1mm crack width and depth).”

Please describe the experimental results for the other two specimens.

(9)  “Page 10, Line 280-293”

“For an intact beam, the first natural frequency was 64.657 Hz. However, as the crack depth grows from 0.5 mm to 2.5 mm…Despite its absolute difference between intact beams 65 Hz and portions of the structure.”

Please write the location of the specific data corresponding to these analyses.

(10) “Page 15, Line 403-405”

“The new crack paths are defined in Table 7. The crack path end points considered not only stepped 0.5 mm. In other words, the tested crack paths include segments with different inclined angles rather than only 45°.”

Please explain the specific defining principles of the new crack path and the basis for the selection of the tilt angle.

(11) “Page 15, Line 412-414”

“Even if the crack has a different orientation angle(such as a1b2c3), the ANN model still gives accurate results with a 1.564 % of the Mean Absolute Percentage Error (MAPE).”

Please explain the basis for the calculation of the data on the mean absolute error percentage.

(12) In this paper, there is a lot of data on crack extension paths, please indicate the specific data locations in the Results and Discussion section.

Comments on the Quality of English Language

The English could be improved to more clearly express the research.

Round 2

Reviewer 2 Report

Comments and Suggestions for Authors

The manuscript was revised following the reviewer's comments. I suggest publication.

Author Response

Thank you for your kind feedback and support. I appreciate your recommendation for publication.

Reviewer 3 Report

Comments and Suggestions for Authors

Manuscript improved quality. It is acceptable.

Author Response

(The authors gave the same response as above.)
